published: 20 August 2021

# Critical Health Literacy in a Pandemic: A Cluster Analysis Among German University Students

Katherina Heinrichs[1]*, Thomas Abel[2], Paula M. Matos Fialho[3], Claudia R. Pischke[3], Heide Busse[4], Claus Wendt[5] and Christiane Stock[1]

[1]Institute of Health and Nursing Science, Charité – Universitätsmedizin Berlin, Corporate Member of Freie Universität Berlin and Humboldt-Universität zu Berlin, Berlin, Germany, [2]Institute of Social and Preventive Medicine, University of Bern, Bern, Switzerland, [3]Institute of Medical Sociology, Centre for Health and Society, Medical Faculty, University of Duesseldorf, Duesseldorf, Germany, [4]Leibniz Institute for Prevention Research and Epidemiology-BIPS, Bremen, Germany, [5]Department Sociology of Health and Healthcare Systems, University of Siegen, Siegen, Germany

**Objectives:** In the COVID-19 pandemic, critical health literacy (CHL-P) has been proposed as a means of addressing issues of complexity, uncertainty, and urgency. Our study aimed to identify CHL-P clusters among university students in Germany and to analyze associations with potential determinants.

**Methods:** In May 2020, students at four German universities participated in the COVID-19 International Student Well-Being Study, an online survey that yielded a non-probabilistic sample of N = 5,021. CHL-P, COVID-19-related knowledge, worries, risk perception, and adherence to protective measures were measured in an online questionnaire with self-constructed items. We conducted a cluster analysis of the five CHL-P items and performed logistic regression analyses.

**Results:** Two CHL-P clusters were identified: high vs. moderate CHL-P. Belonging to the high-CHL-P cluster (31.2% of students) was significantly associated with older age, female/other gender, advanced education, higher levels of parental education, and moderate importance placed on education. In addition, higher levels of knowledge, risk perception and worries, and adherence to protective measures were associated with high CHL-P cluster membership.

**Conclusion:** Students would benefit from educational measures that promote CHL-P at German universities.

Keywords: risk perception, COVID-19, university students, adherence, knowledge, critical health literacy, worries

**Edited by:**
Stéphanie Baggio,
Geneva University Hospitals (HUG),
Switzerland

**Reviewed by:**
Kirsti Riiser,
Oslo Metropolitan University, Norway
Nina Schnyder,
Department of Justice and Home
Affairs of the Canton of Zurich,
Switzerland

**\*Correspondence:**
Katherina Heinrichs
katherina.heinrichs@charite.de

**Citation:**
Heinrichs K, Abel T, Matos Fialho PM,
Pischke CR, Busse H, Wendt C and
Stock C (2021) Critical Health Literacy
in a Pandemic: A Cluster Analysis
Among German University Students.
Int J Public Health 66:1604210.

## INTRODUCTION

In 2020, the SARS-CoV-2/COVID-19 pandemic affected almost every country throughout the world. Depending on the national policy, public life came to a halt in many places. Citizens were confronted with strict protective measures, such as the closing of kindergartens, schools, shops, and restaurants, as well as with further consequences of social distancing, such as working from home while taking care of children and dealing with social isolation. In Germany, university students were additionally confronted with fundamental changes when the summer semester began in May 2020. For instance, lectures and seminars were offered in an online format only, and in-person contact at

universities with other students or teachers was not possible. Students were concerned about not being able to proceed with their education as intended, and some experienced financial hardships, for instance, due to the loss of a job (e.g., in the food-service industry).

While such conditions have continued to cause considerable distress among the students since the pandemic began [1], strict adherence to protective measures–such as social-distancing practices–plays a key role in combating the global health crisis. Young individuals–especially those aged 20–24—are believed to adhere less strictly to social-distancing practices compared with older individuals and hence to significantly contribute to increasing the spread of COVID-19 [2]. Moreover, adherence to COVID-19-related protective measures may be associated with risk perception and mortality rates as well as to knowledge about the nature of the SARS-CoV-2 virus [3, 4].

Furthermore, all citizens have had to cope with high levels of urgency, uncertainty, and unpredictability in major domains of their lives. Additionally, they have had to process a large volume of health information provided by many different sources, some of which are more trustworthy than others [5, 6]. People have been confronted with the task of processing this information and integrating it with personal needs and knowledge insecurities. In addition, they have faced increased responsibility toward their communities in the choice to follow recommended protective measures [5].

The ability to assess, process, and apply health-related information–i.e., health literacy–comprises three levels [7]: 1) basic/functional literacy, such as reading and writing as well as effective functioning in everyday life, 2) communicative/ interactive literacy (i.e., advanced cognitive and social skills required to understand information and to use this information in different situations), and 3) critical literacy, which comprises cognitive and social skills at a higher level required to critically process and to apply this information. Critical health literacy covers three components [8]: 1) to critically analyze health-related information, including its trustworthiness and its applicability to one's own life context, 2) to understand social determinants of health, and 3) to engage in collective action, e.g., to reduce social inequality regarding health, up to a political level.

In the context of the COVID-19 pandemic, the term "critical health literacy in a pandemic" (CHL-P) was introduced in 2020 [6]. The aim was to account for the skills needed in the pandemic, i.e., to deal with an abundance of information and to evaluate the trustworthiness of information before the background of yet growing scientific evidence and political and social uncertainty [5]. Furthermore, it became very obvious during the pandemic how individual and collective actions and responsibilities are intertwined [6]. Therefore, the concept suggests that the acute crisis resulting from a pandemic requires all citizens to recognize and have a basic understanding of the complex issues associated with the pandemic [9]. As a result, CHL-P refers to two main concerns in the pandemic: complexity perception and uncertainty of knowledge [9].

As CHL-P is a new concept, little is known about its distribution within the population. CHL-P can be expected to be associated with other health-related factors, such as general (health) literacy. As health literacy is generally associated with social class in Germany [10], CHL-P can be hypothesized to also be associated with socioeconomic background. In addition, age may play a role because previous research has revealed that levels of health literacy decrease with increasing age in Germany [10]. International research suggests that gender was also found to be associated with health literacy [11]. However, the extent to which CHL-P is linked to adherence to governmental recommendations as well as to factors that could potentially predict such adherence (e.g., COVID-19-related knowledge and risk perception) remains unclear. We expect individuals with higher levels of CHL-P to also be more likely to adhere to governmental recommendations and at the same time to display higher levels of COVID-19-related knowledge and greater risk perception as well as to report more worries.

To examine these hypotheses, we aimed to

1) describe the level of CHL-P among German university students,
2) identify and interpret clusters of CHL-P items,
3) characterize the student composition of each of the emerging clusters in terms of sociodemographic characteristics, and
4) examine the associations of the emerging CHL-P clusters with students' COVID-19 knowledge, risk perception, worries, and adherence to protective measures.

## METHODS

### Setting

Data were collected as part of the "COVID-19 International Student Well-Being Study" (C19 ISWS), a cross-sectional online survey which was conducted in 27 mostly European countries [12]. The study protocol and questionnaire were developed by the coordinating team at the Centre for Population, Family and Health at the University of Antwerp, Belgium (Sarah Van de Velde, Veerle Buffel, Edwin Wouters). The questionnaire is publicly accessible [13]. The items to assess CHL-P were only applied in Germany and Switzerland and are not presented as part of the published C19 ISWS questionnaire.

For the present study, the German data were analyzed. Four German universities took part in the C19 ISWS: Charité–Universitätsmedizin Berlin, the University of Bremen, the University of Siegen, and the Heinrich Heine University Düsseldorf. The questionnaire was designed to assess the psychological impact of the first lockdown in 2020 and was independently translated by two members of the German study team according to the C19 ISWS study protocol [12]. The wording was discussed with all German-speaking C19 ISWS countries until a consensus was achieved. Students in Germany were invited to complete the questionnaire in German or English.

## Sample

Inclusion criteria were 1) enrolment at a participating university as bachelor's, master's, or PhD student, as a candidate for the state examination, or as a participant in other types of programs and 2) a minimum age of 17 years. The aim concerning the sample size was to reach at least 10% of all students at each university.

## Data Collection

The questionnaire was administered during the first lockdown in Germany in response to the COVID-19 pandemic. This lockdown began on 22 March, 2020 and comprised the shutdown of universities and many other public institutions, cultural locations, shops, and restaurants as well as the implementation of social-distancing measures. Classes were taught online in the summer semester of 2020 (beginning on 1 May). Around end of April 2020, some restrictions were lifted, and public places–including playgrounds, museums, and churches–opened again.

In the second half of May 2020, a link to the online survey was sent to all students at the participating universities via emailing lists and/or through notifications on the university webpage as well as through social media and additional websites, including e-learning platforms. A reminder email was sent out after 1 week.

## Data Processing

A data-secure web platform was provided by the University of Antwerp to collect data in all participating countries. Data-protection regulations were followed in all countries, and ethical approval for conducting the study was obtained from the ethics committees at all participating universities. All participants provided informed consent for taking part in the survey. The collection and processing of data were performed with Qualtrics (Qualtrics, Provo, UT, United States).

## Measures

In addition to items assessing sociodemographic characteristics (age, gender, and parents' level of education) and information on the studies (program and importance compared with other activities), the following scales were included in the questionnaire:

### Critical Health Literacy in a Pandemic

Items were formulated to assess the two main aspects of CHL-P–complexity perception and uncertainty of knowledge on public health actions (e.g., "The biggest problem in this pandemic is with the high-risk groups (e.g., 65+; people with chronic health problems); consequently, the behavioral restrictions should apply only to them"). They were developed in an interdisciplinary group of public health experts via three rounds of expert feedback and were pre-tested among young adults in German, French, and English by the Swiss C19 ISWS coordinating team [9].

Five response options per item were provided and adapted to the respective content and therefore differed across items (e.g., ranging from "fully agree" to "fully disagree"). For presentation purposes in this paper, we altered the numbering of the items. In the questionnaire, they were presented in the original order [9]. A dichotomization of each item that distinguished between low and

high levels of CHL-P was proposed on a theoretical basis by Abel and Benkert [9]. According to the authors, high CHL-P is defined as:

- Answers "rather strong," "neutral," "rather weak" to the first item ("How would you rate the current scientific knowledge on COVID-19 available to guide political decisions in Germany?"),
- Answers "neutral," "rather important," "very important" to the second item ("How important is it for you to understand the often-different interests and motivations among the key players in this crisis?"), and
- Answers "neutral," "rather disagree," "fully disagree" to the third to fifth items ("The challenges in this crisis are simple, and decision-making is fairly straightforward."; "The behavioral restrictions should apply to high-risk groups only.", "Individuals are equally affected, independent of their social class or status").

### Knowledge of SARS-CoV-2/COVID-19

To obtain information on the students' knowledge of SARS-CoV-2/COVID-19, eight items were used (items Q42a-h in the questionnaire, e.g., "You can have the virus without any symptoms."). The response options were "true," "false," and "don't know." Based on the C19 ISWS protocol, answers were dichotomized into groups of correct answers (value = 1, correct answer) and wrong answers, the latter of which included the "false" and "don't-know" options (values = 0, incorrect answer). Subsequently, a sum score ranging from 0 (all incorrect) to 8 (all correct) was computed (no missing values accepted) and dichotomized at its median.

### Risk Perception, Worries, and Adherence

The students were asked to rate the subjective likeliness of a COVID-19 infection with one item (item Q29b in the questionnaire, "In your opinion, how likely are you to get infected by COVID-19?") in order to measure their risk perception. Their worries concerning COVID-19 were assessed with five items (items Q30b (2 items), Q31a-b (2 items), and Q32 in the questionnaire, e.g., "How worried are you to get infected with COVID-19?" or "How worried are you that doctors and hospitals will not have adequate medical supplies to handle the COVID-19 outbreak?"). An additional question referred to how strict the students had adhered to the protective measures implemented by the government at the time of the survey (item Q34 in the questionnaire).

The response options for rating subjective likeliness, worries, and adherence ranged from zero ("very unlikely," "not worried at all," and "totally not," respectively) to ten ("very likely," "very worried," and "very strictly," respectively). For the statistical analyses, the items that assessed risk perception and adherence to protective measures were dichotomized at their median. The five items that quantified worries were added (no missing values accepted) in line with Tasso et al. [1], and the sum score (ranging from 0 (not worried at all) to 50 (very worried)) was dichotomized at the median.

**TABLE 1 |** Sample characteristics, *n* = 5,021; COVID-19 International Student Well-Being Study, Germany, 2020.

| Variable | x̄ (SD) or *n*; % |
|---|---|
| Age (years)[a] | 24.4 (5.0) |
| Gender[a] | |
| Female | 3,485; 69.4 |
| Male | 1,478; 29.4 |
| Other | 58; 1.2 |
| Type of program[a] | |
| Bachelor's | 2,700; 53.8 |
| Master's | 1,138; 22.7 |
| PhD | 234; 4.7 |
| State examination (e.g., medicine, law) | 910; 18.1 |
| Other | 39; 0.8 |
| Parents' level of education | |
| High[b] | 2,501; 49.8 |
| Middle[c] | 1,079; 21.5 |
| Low[d] | 1,229; 24.5 |
| Missing values | 212; 4.2 |

x̄, mean; SD, standard deviation.
[a]No missing values.
[b]At least one parent with higher education.
[c]At least one parent with secondary-school degree.
[d]Both parents with no secondary-school degree.

## Data Analysis

All analyses were conducted with IBM SPSS Statistics 27. In the first step, the variables were descriptively analyzed.

As the items that assessed CHL-P displayed low correlation with one another and the Kaiser–Meyer–Olkin (KMO) measure was rather low (KMO = 0.503), a factor analysis was not indicated. Therefore, the items could not be treated as one scale, for example, to compute a sum score. In order to divide our sample into CHL-P-related subgroups and to identify groups with consistent response patterns across all five items, a hierarchical cluster analysis was performed. We employed the Euclidean distance measure for continuous variables [14], which is also commonly used for Likert scales [15], with between groups linkage and performed a z-transformation of the variables. The two-cluster solution was based on the dendrogram plot.

In order to test the associations of the resulting CHL-P clusters with knowledge, risk perception, worries, and adherence as well as sociodemographic characteristics and information on the participants' studies, logistic regressions were performed to calculate odds ratios (ORs) with 95% confidence intervals (CIs). As we stated above, age, gender, and socioeconomic background might interact with CHL-P, so we adjusted for age, sex, and parents' level of education. We only used complete datasets without missing values leading to samples of 4,653 cases or 4,622 cases, depending on the respective regression analysis.

We tested the possible predictors of risk perception, worries, and adherence to protective measures (7 items in total) for collinearity in our model. The variance inflation factors (VIFs) were under 3.4, which indicates that collinearity was moderate and did not need to be considered a severe problem in our regression model.

## RESULTS

## Sample Description

A total of 5,021 students participated in the study. The aim to reach 10% of the overall population of the four participating universities was achieved for three of them, where the response rate was approximately 10–11%. One university mainly recruited via social media, and only at one of its five faculties (Medical Faculty), students received additional email invitations. About 2% of all students at this university participated in the study, whereas at the Medical Faculty, approximately 17% took part. **Table 1** presents the sample characteristics. The mean age was 24.4 years, and over two-thirds of the participants were female. Over 50% were enrolled in a bachelor's program, and nearly one-quarter in a master's program. Nearly one in five participants were enrolled in another type of program: state examination, which is required to become a physician, among others, in Germany. The subgroup studying medicine was probably overrepresented in our sample as the data assessment predominantly took place at two medical faculties (Berlin and Düsseldorf). Almost half of the students reported high levels of education for both parents (i.e., a university degree or the equivalent), whereas nearly one-quarter of the sample stated that both their parents had no secondary-school degree (i.e., a university-admission qualification).

In **Table 2**, the numbers and percentages of students with a high level of CHL-P are presented. According to the questions that assessed CHL-P and their theoretical dichotomization, the majority of study participants reported a high level of CHL-P (84.1–92.4%), except for in one question, which split the sample in half: A total of 53.1% showed a high level of CHL-P by disagreeing with the item "Independent of their social class or status, individuals are equally affected by the current pandemic."

A total of 54.0% of the sample were classified to have a high level of knowledge of COVID-19—that is, they answered six to eight of the eight items correctly, while the mean of correct answers was 5.5 (see **Table 3**). The mean for risk perception–that is, the subjective likeliness of becoming infected–was 4.4 (on a scale of 1–10). The mean of the summed items that assessed worries concerning COVID-19 was 24.7 out of 50 (Cronbach's alpha: α = 0.81). Participants were mostly worried that someone from their personal network would become severely ill from a COVID-19 infection (6.9 out of 10) or would become infected at all with COVID-19 (6.5/10). The equivalent worries concerning oneself were much lower, and the worry that there might not be enough medical supplies was 4.9/10. Students reported adhering fairly strictly to protective measures (8.0/10).

## The CHL-P Clusters

The hierarchical cluster analysis and the evaluation of the dendrogram plot led to two clusters of CHL-P. The larger cluster comprised around two-thirds of the sample and could be interpreted as a group with a moderate level of CHL-P, whereas the smaller group (31.2%) could be considered to show a high level of CHL-P. **Figure 1** displays the answers to the five CHL-P items according to the two CHL-P clusters. The first item did not generate great differences between the two

**TABLE 2 |** Levels of critical health literacy in a pandemic (CHL-P), items shortened, n = 5,021; COVID-19 International Student Well-Being Study, Germany, 2020.

| CHL-P items (5-tier answer format) | Answer format | High CHL-P; n; % | Low CHL-P; n; % | Missing values; n; % |
|---|---|---|---|---|
| How would you rate the current scientific knowledge on COVID-19 available to guide political decisions in Germany? | very strong to very weak[a] | 4,272; 85.1 | 599; 11.9 | 150; 3.0 |
| How important is it for you to understand the often-different interests and motivations among the key players in this crisis? | very important to not important[b] | 4,639; 92.4 | 226; 4.5 | 156; 3.1 |
| Overall, the challenges in this COVID-19 crisis are simple, and decision-making is fairly straightforward | fully agree to fully disagree[c] | 4,221; 84.1 | 648; 12.9 | 152; 3.0 |
| The biggest problem in this pandemic is with the high-risk groups; consequently, the behavioral restrictions should apply only to them | fully agree to fully disagree[c] | 4,296; 85.6 | 567; 11.3 | 158; 3.1 |
| Independent of their social class or status, individuals are equally affected by the current pandemic | fully agree to fully disagree[c] | 2,667; 53.1 | 2,197; 43.8 | 157; 3.1 |

[a-c]High levels of CHL-P theoretically defined by Abel and Benkert [9]:
[a]Three middle positions of 5-tier answer format.
[b]Neutral and agreement.
[c]neutral and disagreement.

**TABLE 3 |** COVID-19-related knowledge, risk perception, worries, and adherence to protective measures, n = 5,021; COVID-19 International Student Well-Being Study, Germany, 2020.

| Variable | x̄ (SD) or n; % |
|---|---|
| Knowledge of SARS-CoV-2/COVID-19 | |
| Sum score (on a scale of 0 (all incorrect)—8 (all correct)) | 5.5 (1.1) |
| Missing values | 126; 2.5 |
| Risk perception (on a scale of 1–10) | |
| How likely are you to get infected? | 4.4 (2.4) |
| Missing values | 81; 1.6 |
| Worries (on a scale of 1–10) | |
| How worried are you … | |
| … to get infected with COVID-19? | 3.6 (2.6) |
| … that you will get severely ill from a COVID-19 infection? | 2.8 (2.7) |
| … that anyone from your personal network will get infected with COVID-19? | 6.5 (2.7) |
| … that anyone from your personal network will get severely ill from a COVID-19 infection? | 6.9 (2.8) |
| … that doctors and hospitals will not have sufficient supplies to handle the COVID-19 outbreak? | 4.9 (3.1) |
| Worries about COVID-19 (sum score on a scale of 1–50) | 24.7 (10.6) |
| Missing values | 88; 1.8 |
| Adherence (on a scale of 1–10) | |
| To what degree do you adhere to the COVID-19 measures currently implemented by the government? | 8.0 (1.8) |
| Missing values | 54; 1.1 |

x̄, mean; SD, standard deviation.

groups. Whereas almost all students with a high level of CHL-P considered it important to understand the interests and motivations among key players (e.g., the government, political parties, employer organizations, unions, health authorities), only 72.7% of students with a moderate level of CHL-P had similar feelings. The next two items–which stated that the challenges during the pandemic had been simple and that restrictions should only have applied to high-risk groups–led to homogenous answers among the high-CHL-P cluster, with almost all members disagreeing, whereas among the moderate-CHL-P cluster, 59.8 and 73.1% disagreed with each statement, respectively. This difference became even more pronounced when examining the last item: 99.6% of the high-CHL-P cluster disagreed with the statement that all individuals had been equally affected by the pandemic, whereas among the moderate-CHL-P cluster, only 28.0% disagreed.

## CHL-P Clusters and Sociodemographic and Study-Related Correlates

Results of unadjusted and adjusted models are shown in **Table 4**. In the text, we present the results of the adjusted models. The unadjusted models delivered comparable results. Students above the age of 25 had about 1.5-fold odds of belonging to the high-CHL-P cluster compared with students under 25 (OR = 1.48; 95% CI: 1.30–1.69). Moreover, female students and students with a non-binary gender identity were more likely to belong to the high-CHL-P cluster than were males (OR = 1.20; 95% CI: 1.05–1.38 and OR = 2.45; 95% CI: 1.36–4.39, respectively). Compared with students at the beginning of their training (a bachelor's program), students in a master's or PhD program had greater odds of belonging to the high-CHL-P cluster (OR = 1.31; 95% CI: 1.12–1.54 and OR = 1.45; 95% CI: 1.08–1.95, respectively), and the odds were even greater among students

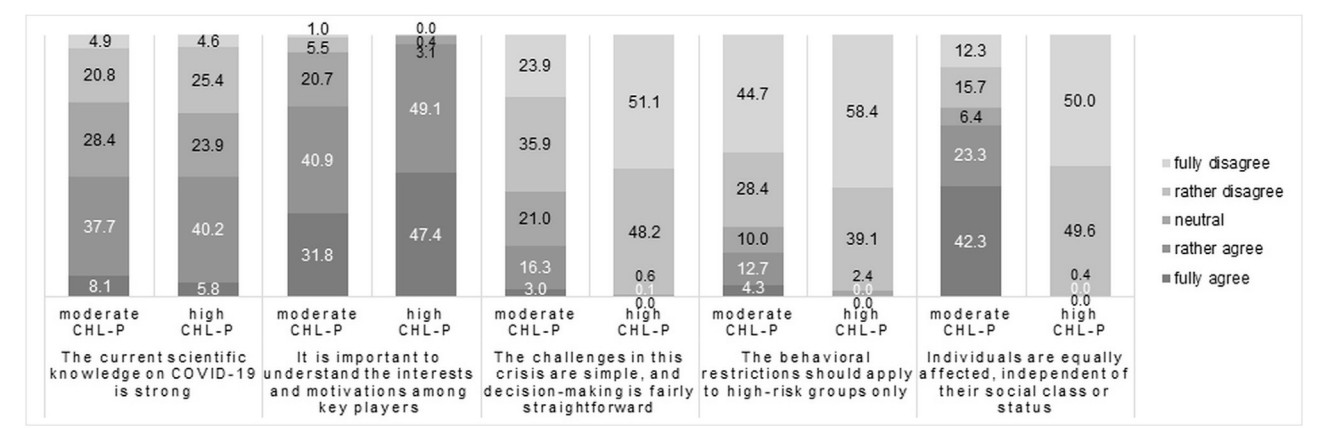

**FIGURE 1 |** Responses to critical-health-literacy-in-a-pandemic (CHL-P) items by computed CHL-P clusters (moderate vs. high CHL-P); items shortened, responses in %; COVID-19 International Student Well-Being Study, Germany, 2020.

**TABLE 4 |** Odds ratios and 95% confidence intervals for the high-critical-health-literacy-in-a-pandemic cluster according to possible determinants; COVID-19 International Student Well-Being Study, Germany, 2020.

| Possible determinants | Unadjusted model | | | Adjusted model[a] | | |
|---|---|---|---|---|---|---|
| | OR | 95% CI | *p* | OR | 95% CI | *p* |
| Age | | | | | | |
| <25 | Ref. | — | — | Ref. | — | — |
| ≥25 | 1.41 | 1.24–1.60 | <0.001 | 1.48 | 1.30–1.69 | <0.001 |
| Gender | | | | | | |
| Male | Ref. | — | — | Ref. | — | — |
| Female | 1.16 | 1.01–1.33 | 0.032 | 1.20 | 1.05–1.38 | 0.009 |
| Other | 2.30 | 1.33–3.97 | 0.003 | 2.45 | 1.36–4.39 | 0.003 |
| Type of program | | | | | | |
| Bachelor's | Ref. | — | — | Ref. | — | — |
| Master's | 1.43 | 1.23–1.66 | <0.001 | 1.31 | 1.12–1.54 | 0.001 |
| PhD | 1.74 | 1.31–2.31 | <0.001 | 1.45 | 1.08–1.95 | 0.013 |
| State examination (e.g., medicine, law | 1.96 | 1.67–2.30 | <0.001 | 1.78 | 1.51–2.10 | <0.001 |
| Parents' level of education | | | | | | |
| Low | Ref. | — | — | Ref. | — | — |
| Medium | 1.05 | 0.87–1.27 | 0.586 | 1.11 | 0.92–1.34 | 0.294 |
| High | 1.64 | 1.41–1.92 | <0.001 | 1.73 | 1.48–2.02 | <0.001 |
| Importance of studies | | | | | | |
| More important | Ref. | — | — | Ref. | — | — |
| Equally important | 1.23 | 1.08–1.40 | 0.002 | 1.25 | 1.10–1.42 | 0.001 |
| Less important | 1.08 | 0.83–1.41 | 0.566 | 1.06 | 0.81–1.39 | 0.673 |

OR, odds ratio; CI, confidence interval.
[a]Age adjusted for gender and parents' level of education; gender adjusted for age and parents' level of education; program type and importance of studies adjusted for age, gender, and parents' level of education; parents' education adjusted for age and gender.

working toward a state examination, which included medical students (OR = 1.78; 95% CI: 1.51–2.10). Students whose parents had higher levels of education were also more likely to belong to the high-CHL-P cluster than were those whose parents had lower levels of education (OR = 1.73; 95% CI: 1.48–2.02).

The students were also asked to rate the importance of their studies (i.e., university education) compared with their other activities. Over 50% rated the importance of studies and other activities as being equal, and over one-third rated studies as being more important. Compared with the latter group, the students who considered their education to be equally important as other

activities had elevated odds of belonging to the high-CHL-P cluster (OR = 1.25; 95% CI: 1.10–1.42).

## CHL-P Clusters and Knowledge, Risk Perception, Worries, and Adherence

The associations between high CHL-P and knowledge, risk perception, worries, and adherence to protective measures are displayed in **table 5**. In the text, we present the results of the adjusted models. The unadjusted models delivered comparable results. Students with a higher level of knowledge of COVID-19

**TABLE 5 |** Odds ratios and 95% confidence intervals for the high-critical-health-literacy-in-a-pandemic cluster according to possible determinants; COVID-19 International Student Well-Being Study, Germany, 2020.

| Possible determinants | Unadjusted model | | | Adjusted for age, gender, and parents' level of education | | |
|---|---|---|---|---|---|---|
| | OR | 95% CI | p | OR | 95% CI | p |
| Knowledge | | | | | | |
| Lower level | Ref. | — | — | Ref. | — | — |
| Higher level | 1.33 | 1.17–1.50 | <0.001 | 1.32 | 1.16–1.50 | <0.001 |
| Risk perception | | | | | | |
| Unlikely to infected | Ref. | — | — | Ref. | — | — |
| Likely to get infected | 1.32 | 1.16–1.49 | <0.001 | 1.26 | 1.11–1.43 | <0.001 |
| Worries | | | | | | |
| Lower level | Ref. | — | — | Ref. | — | — |
| Higher level | 1.26 | 1.11–1.42 | <0.001 | 1.25 | 1.10–1.41 | 0.001 |
| Adherence | | | | | | |
| Less adherent to measures | Ref. | — | — | Ref. | — | — |
| More adherent to measures | 1.27 | 1.12–1.43 | <0.001 | 1.25 | 1.10–1.41 | 0.001 |

*OR, odds ratio; all possible determinants dichotomized at median; CI, confidence interval.*

had elevated odds (OR = 1.32; 95% CI 1.16–1.50) of belonging to the high-CHL-P cluster compared with participants with a lower level of knowledge of COVID-19. Furthermore, the conviction that is was likely to become infected was associated with a high level of CHL-P (OR = 1.26; 95% CI 1.11–1.43). Participants who reported a higher level of worries about COVID-19 also had elevated odds of belonging to the high-CHL-P cluster (OR = 1.25; 95% CI 1.10–1.41), as did students who adhered more strictly to the protective measures that had been implemented by the government at the time of the survey (OR = 1.25; 95% CI 1.10–1.41).

# DISCUSSION

This study is one of the first to investigate the newly introduced concept of CHL-P among university students in Germany. Our results provide first insights into the distribution of high vs. moderate levels of CHL-P and show that university students in Germany tend to report a high level of CHL-P. Further, our findings help in better understanding additional characteristics of students with a high level of CHL-P: Cluster membership (high vs. moderate CHL-P) was associated with several sociodemographic characteristics as well as COVID-19-related knowledge, worries, risk perception, and adherence.

Regarding the first aim of the study, a high level of CHL-P was reflected in more than 80% of responses to four of the five items among students from the theoretically defined high-CHL-P group. Scores on the fifth item, however, suggest that about half of the survey participants were unaware of the relationship between being affected by the pandemic and social class. These results are in line with the C19 ISWS results from a sample of Swiss students [9] when it comes to the response pattern for each of the items, which indicates that country differences do not seem to play a vital role in the overall response pattern. However, the level of CHL-P was somewhat lower in our sample than in Switzerland [9], where more than 90% of respondents scored high in CHL-P for at least three of the

five items, more than 85% for a fourth item, and more than 70% for the social-class-related item.

To address our second aim, we studied clusters of students with consistent response patterns across all five items and could identify two clusters. The answers among clusters 1 and 2 confirmed the theoretical definition of high and low CHL-P given by Abel and Benkert [9]. It became clear that cluster 1 comprised students with high CHL-P responses across (almost) all five items. This group included only 31% percent of the German student sample. The remaining 69% showed a moderate level of CHL-P, which might seem a high percentage, but is supported by other research, which has demonstrated that about 50% of the adult population in Germany displays problematic or inadequate levels of COVID-19-related health literacy [16]. However, the tool to measure COVID-19-related health literacy assessed how the participants accessed, understood, appraised, and applied health-related information during the COVID-19 pandemic and thus, differed from the concept of CHL-P, and percentages cannot be directly compared.

Concerning our third aim, we identified several sociodemographic and study-related characteristics associated with high CHL-P, namely higher age, female or diverse gender identity, an advanced higher study program, high levels of parental education, and a moderate importance of studies. These results indicate that CHL-P might increase with increasing age in young adulthood due to the development of critical-thinking abilities. A previous meta-analysis showed that students from different health professions improved their critical thinking during the course of their studies [17]. Our study also indicates that CHL-P increases with progress in one's own education (e.g., transitioning from bachelor's to master's studies). Similar effects were demonstrated in the Swiss C19 ISWS sample, in which students who were enrolled in different programs reported different levels of awareness of the relationship between being affected by the pandemic and social class [9]. This finding reflects the important role that education plays for the level of CHL-P, which has similarly been demonstrated for health literacy in general [18].

As one may expect, students with well-educated parents were more likely to belong to the high-CHL-P cluster, thus indicating that educational resources in the family may be an underlying determinant of high CHL-P. CHL-P clusters also differed by gender identity, with lower CHL-P among male students. This finding is in line with the data from Switzerland, which showed that male students were more likely to uncritically rate most of the CHL-P items [9], and with the European health literacy survey data, which revealed slightly lower general health literacy levels among men in Europe [18].

Regarding our fourth study aim, students reported high levels of adherence to protective measures on average. With respect to their risk perception, their worries concerning their social network exceeded their worries that they themselves could become infected or ill, which indicates that they seemed to have followed the restrictions and rules to protect mainly others as opposed to themselves. This result was also found among a Canadian sample of adolescents and young adults, who perceived lower risk for themselves than for their relatives and were more likely to adhere to the protective measures the higher their risk perception was [19]. Concerning COVID-19-related knowledge, high levels were noted for only about half of the student sample. In agreement with our findings, mixed results also exist concerning knowledge about SARS-CoV-2/COVID-19, which ranges from moderate [20, 21] to high levels among students of the life sciences [22]. When studying the associations of all of these factors with belonging to the high-CHL-P cluster, we found consistently positive associations for adherence to governmental recommendations, COVID-19-related worries, and knowledge level with this high-CHL-P cluster. However, in our sample, less than one in three students belonged to the high-CHL-P cluster. Due to a probable overrepresentation of medical students in the sample, the proportion of high-CHL-P students can be assumed to be even lower in the general German student population. This observation stresses the importance of promoting CHL-P among university students in Germany and the role universities can play as educational institutions.

## Strengths and Limitations

During the first lockdown resulting from the COVID-19 pandemic, the C19 ISWS research team and CHL-P scale developers reacted quickly in designing this innovative study, which was one of the first to assess CHL-P at an international level among university students during the COVID-19 pandemic. Pioneering steps were taken from the general concept of (critical) health literacy to the concept of CHL-P, which is critical and urgent in the current situation.

The study was realized not only in Germany, but in 27 countries. We reached thousands of participants and thus, the targeted 10% of students at three of our four sites. Thereby, we gained detailed insights into health perceptions and behaviors that are linked with the COVID-19 pandemic and the differing lockdown experiences. Furthermore, we investigated the newly formulated concept of CHL-P, which might help in providing a better understanding of and promoting health-related behavior among young adults.

Nonetheless, our study has some limitations. First, as it is cross-sectional, we cannot draw conclusions concerning causal relationships between CHL-P and the other variables that we investigated. Second, our results apply to the first lockdown in Germany, during the spring of 2020. Perceptions and behaviors may have changed since then as the pandemic has proven to be unpredictable. For instance, COVID-19-related knowledge may have improved with time, and the worry about becoming infected with COVID-19 may have increased among students in early 2021, when mutations emerged that went hand in hand with an increased risk of suffering from a severe progression of COVID-19 for younger people than would have been the case with the original SARS-CoV-2 virus. However, the beginning of the vaccination campaign around the same time might have had a comforting effect on the German population. Nevertheless, even if certain variables have changed with time, this change should have no effect on the observed associations between CHL-P clusters and our outcome variables. In order to observe changes over time, a second assessment is needed, which would ideally include more study centers to increase representativity.

Third, due to the limited response rate of 10–11%, a selection bias towards students more interested in health and well-being and thus potentially of higher level of CHL-P cannot be ruled out. Additionally, medical students were probably overrepresented in our sample which therefore must be considered as non-representative for the overall population of university students in Germany. This might be due to the applied recruitment strategies, which differed across the participating institutions. However, we performed regression analyses adjusted for the study field "health" (vs. rest) which delivered comparable results. Only women and PhD students had no longer significantly higher odds of belonging to the high-CHL-P cluster. Knowledge and CHL-P might be assumed to be lower in a representative sample of students in Germany. This observation highlights the relevance of promoting CHL-P and COVID-19-related knowledge among university students.

Fourth, we collected data using only self-reports. The students rated their own perceptions and behaviors, and only their COVID-19-related knowledge was tested. The effects of social desirability cannot be ruled out, especially in terms of adherence, and could have led to an overestimation of the adherence to protective measures. Future research might consider searching for a more objective way to assess the data or could include a scale that measures social desirability in the questionnaire for statistical adjustment.

Fifth, we did not use validated scales to assess our variables because there was no opportunity to pre-test the new scales due to time pressure. Ceiling effects might have played a role, especially concerning CHL-P, as the majority of the sample had a high level of CHL-P for four of the five items. This observation leads us to believe that the items were too easy in a statistical sense or that the differentiation of answers that indicated a high level of CHL-P was not narrow enough. It would thus be wise to validate and potentially also adapt the scales for measuring CHL-P and worries. Further, some of the items that assess the other COVID-19-related variables, especially the scale measuring

knowledge, would require adaptation before subsequent use because of the ever-changing situation and knowledge concerning COVID-19. Overall, our findings should be interpreted in the context of lacking validation and for the time point of data collection only. Our data collection took place during the first lockdown in spring 2020, and our results are not fully generalizable beyond that time span. Nevertheless, they give valuable first insight into the concept of CHL-P.

## Conclusion

Our results suggest that students in Germany have differing levels of CHL-P and can be categorized into different clusters. Only one-third of respondents reported a high level of CHL-P, which indicates that the majority of students might benefit from educational measures that promote CHL-P in German universities. Additionally, when aiming to increase CHL-P, it would be wise to take further factors into account, such as the importance students place on their university education, COVID-19-related knowledge, risk perception, worries about the potential for infection, and adherence to protective measures. It is not yet known how these variables interact, but it might be worthwhile to focus on as many of them as is feasible, e.g., in workshops or projects, (digital) information campaigns or individual counseling. Furthermore, the correlates indicate that vulnerable groups–such as young and/or male students, students at the beginning of their university studies, and students with low-educated parents–should be targeted by strategies aimed at promoting CHL-P. With a higher level of CHL-P, students might be able to process information on the pandemic and evaluate their trustworthiness on a more advanced level which could help them to understand the interrelationships between individual and collective actions and responsibilities as well as the determinants of social inequalities related to the pandemic. As a result, they might be able to recognize the complexity of the pandemic situation as well as the uncertainty of knowledge and act accordingly at both individual and collective levels.

## ETHICS STATEMENT

The studies involving human participants were reviewed and approved by the Ethics Committees of all four participating universities (Charité–Universitätsmedizin Berlin, University of Bremen (protocol code 2020-04-EILV, dated May 4, 2020), Heinrich Heine University Düsseldorf (protocol code 2020-958, dated May 5, 2020) and the University of Siegen (protocol code ER 08/2020, dated May 7, 2020)). Written informed consent from the participants' legal guardian/next of kin was not required to participate in this study in accordance with the national legislation and the institutional requirements.

## AUTHOR CONTRIBUTIONS

Conceptualization: KH, TA, CP, HB, CW, and CS; data assessment: TA, CP, HB, CW, and CS; data processing and analyses: KH and CS; interpretation of results: all authors; drafting: KH and CS; revising the work critically for important intellectual content: all authors; final approval of the version to be published and agreement to be accountable for all aspects of the work: all authors.

## CONFLICT OF INTEREST

The authors declare that the research was conducted in the absence of any commercial or financial relationships that could be construed as a potential conflict of interest.

## ACKNOWLEDGMENTS

We would like to thank all participants who took part in this survey.

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
