## [Reviewer comments · International Journal of Public Health]

Peer Review Report

Review Report on Critical health literacy in a pandemic: a cluster analysis among German university students

Original Article, Int J Public Health

Reviewer: Kirsti Riiser

Submitted on: 13 Jun 2021

Article DOI: 10.3389/ijph.2021.1604210

EVALUATION

Q 1 Please provide your detailed review report to the authors. The editors prefer to receive your review structured in major and minor comments. Please consider in your review the methods (statistical methods valid and correctly applied (e.g. sample size, choice of test), is the study replicable based on the method description?), results, data interpretation and references. If there are any objective errors, or if the conclusions are not supported, you should detail your concerns.

This interesting and informative article on critical health literacy among university students in the early phase of the COVID-19 pandemic is an important contribution to the field of health literacy research. Young people are more likely to socialize in groups and it is obviously important that they acquire, assess and apply pandemic-related health information and adapt their behavior to avoid getting infected and infecting others. Although the basic protective measures may seem easy to understand, the rapidly shifting regulations and massive information flow may be challenging to comprehend and assess, even for university students. I have only some comments and suggested revisions to the article.

Major comments:

- The intro as well as the discussion section seem primarily to deal with information appraisal and less with the understanding of social determinants and engagement in collective action as defined by Nutbeam and elaborated on by e.g. Chinn. Individual contribution to community outcomes is utterly relevant in the context of a pandemic. The authors refer to Nutbeam in the introduction and should thus use the opportunity to discuss their results more broadly related to different dimensions of CHL and the actual meaning and implications of the results in the context of the pandemic.
- The instrument applied to measure the primary outcome, CHL-P, is described as measuring “complexity perception” in the reference given (Abel & Benkert, 2020). It may be obvious to experts that complexity perception and CHL-P are identical constructs, but since the instrument is recently developed and unfamiliar to most readers, I urge the authors to explain the link between the two constructs more thoroughly.

Minor comments:

- P1, line 13: The authors should name both of the two clusters identified.
- P2, line 31: The sentence should read “students were concerned about not being able to...”
- P3, line 72: The study design should be described initially in the methods section.
- P3, line 73 and onwards: The methods section is not structured with subsections for setting and sample. It is easier to get an overview if the participants are described under a designated section with a heading. The same goes for data collection. Also, did the authors estimate a sample size prior to study start?
- P4, line 109: I may have missed it, but I could not find the CHL-P scale in the published questionnaire. Was it added just for the German students? If so, this should be described more clearly.
- P4, line 113: I recommend that the authors add some more information about the CHL-P instrument (ref. major comment), e.g. how this pandemic-specific instrument (the items) relates to the overall concept of CHL.
- P4, line 128: How was the “worries”-scale calculated? Min-max-values?
- P4, line 124: How was the sum score calculated? Min-max values?
- P5, line 155: There is no information on handling of missing data.
- P6, line 321: Instead of just repeating the aims, I recommend that the discussion section starts with a short summary of the results with reference to the aims.

P7, line 282: I believe the authors could answer this question by doing a sub-group analysis by withdrawing the state examination/medical students from the total group and compare them with the state examination students. Was this considered?

Q 2 Please summarize the main findings of the study.

This study investigated pandemic-related critical health literacy (CHL-P) among German university students and identified two clusters of students, one high CHL-P cluster and one moderate CHL-cluster of which the latter cluster was the largest, consisting of more than two thirds of the sample. Belonging to the high CHL-P cluster was associated with older age, being female, more advanced education, having parents with higher level of education, and moderate importance placed on education. The odds for belonging to the high CHL-P was greater for students having good knowledge of COVID-19. Moreover being worried about COVID-19, having greater risk perception and stricter adherence to protective measures was associated with a high level of CHL-P.

Q 3 Please highlight the limitations and strengths.

The main limitations of the study are the overrepresentation of medical students in the sample as well as the use of non-validated instruments. The main strength is the timing of the study, that such efforts were made to collect health literacy data from the student population during the COVID-19 pandemic. This may help universities and authorities to plan for interventions aiming to increase students' critical health literacy to be better prepared for future pandemics.

PLEASE COMMENT

Q 4 Is the title appropriate, concise, attractive?

Yes

Q 5 Are the keywords appropriate?

Yes

Q 6 Is the English language of sufficient quality?

Yes

Q 7 Is the quality of the figures and tables satisfactory?

Yes.

Q 8 Does the reference list cover the relevant literature adequately and in an unbiased manner?)

Yes

QUALITY ASSESSMENT

Q 9 Originality

Q 10 Rigor

Q 11 Significance to the field

Q 12 Interest to a general audience

Q 13 → Quality of the writing

Q 14 → Overall scientific quality of the study

REVISION LEVEL

Q 15 → Please take a decision based on your comments:

Minor revisions.

---

## [Reviewer comments · International Journal of Public Health]

Peer Review Report

Review Report on Critical health literacy in a pandemic: a cluster analysis among German university students

Original Article, Int J Public Health

Reviewer: Nina Schnyder

Submitted on: 14 Jun 2021

Article DOI: 10.3389/ijph.2021.1604210

EVALUATION

Q 1 Please provide your detailed review report to the authors. The editors prefer to receive your review structured in major and minor comments. Please consider in your review the methods (statistical methods valid and correctly applied (e.g. sample size, choice of test), is the study replicable based on the method description?), results, data interpretation and references. If there are any objective errors, or if the conclusions are not supported, you should detail your concerns.

See attachment

Q 2 Please summarize the main findings of the study.

The authors analysed the distribution of "critical health literacy during a pandemic" (CHL-P), a new concept but not dissimilar to "health literacy", in a student sample. They found that one-third of students clustered in the high CHL-P group. However, from a statistical point of view, it is somewhat unclear how the authors have arrived at this conclusion.

Q 3 Please highlight the limitations and strengths.

Strength and limitation: large sample size but unclear how representative the sample is for a general student population.

Limitation: none of the used measures were validated.

Strength and limitation: CHL-P seems to be an important concept for public health. However, it is unclear what its advantages are over traditional "health literacy".

Potential limitation regarding the statistical methods that have been used: it is questionable whether the used statistical tests are appropriate.

Limitation: a response rate has neither been reported (results) nor commented (discussion).

PLEASE COMMENT

Q 4 Is the title appropriate, concise, attractive?

Yes.

Q 5 Are the keywords appropriate?

Yes.

Q 6 Is the English language of sufficient quality?

As much as I can judge as a non-native English speaker myself: yes.

Q 7 Is the quality of the figures and tables satisfactory?

Yes.

Q 8 Does the reference list cover the relevant literature adequately and in an unbiased manner?)

This is difficult to answer with the current surge of new literature appearing every day.

QUALITY ASSESSMENT

Q 9 Originality

Q 10 Rigor

Q 11 Significance to the field

Q 12 Interest to a general audience

Q 13 Quality of the writing

Q 14 Overall scientific quality of the study

REVISION LEVEL

Q 15 Please take a decision based on your comments:

Major revisions.